# Zfra Overrides WWOX in Suppressing the Progression of Neurodegeneration

**DOI:** 10.3390/ijms25063507

**Published:** 2024-03-20

**Authors:** Yu-An Chen, Tsung-Yun Liu, Kuan-Yu Wen, Che-Yu Hsu, Chun-I Sze, Nan-Shan Chang

**Affiliations:** 1Institute of Molecular Medicine, College of Medicine, National Cheng Kung University, Tainan 70101, Taiwan; momofish0716@gmail.com (Y.-A.C.); s91009s91008@gmail.com (T.-Y.L.); wen840329@gmail.com (K.-Y.W.); bryanhsuks@gmail.com (C.-Y.H.); 2Department of Cell Biology and Anatomy, College of Medicine, National Cheng Kung University, Tainan 70101, Taiwan; chuni.sze@gmail.com; 3Graduate Institute of Biomedical Sciences, College of Medicine, China Medical University, Taichung 40402, Taiwan; 4Department of Neurochemistry, New York State Institute for Basic Research in Developmental Disabilities, New York, NY 10314, USA

**Keywords:** tumor suppressor, WWOX, Zfra, Hyal-2+ Z cell, cancer, Alzheimer’s disease

## Abstract

We reported that a 31-amino-acid Zfra protein (zinc finger-like protein that regulates apoptosis) blocks neurodegeneration and cancer growth. Zfra binds WW domain-containing oxidoreductase (WWOX) to both *N*- and *C*-termini, which leads to accelerated WWOX degradation. WWOX limits the progression of neurodegeneration such as Alzheimer’s disease (AD) by binding tau and tau-hyperphosphorylating enzymes. Similarly, Zfra binds many protein targets and accelerates their degradation independently of ubiquitination. Furthermore, Zfra4-10 peptide strongly prevents the progression of AD-like symptoms in triple-transgenic (3xTg) mice during aging. Zfra4-10 peptide restores memory loss in 9-month-old 3xTg mice by blocking the aggregation of a protein cascade, including TPC6AΔ, TIAF1, and SH3GLB2, by causing aggregation of tau and amyloid β. Zfra4-10 also suppresses inflammatory NF-κB activation. Zfra-activated Hyal-2+ CD3- CD19- Z cells in the spleen, via Hyal-2/WWOX/Smad4 signaling, are potent in cancer suppression. In this perspective review, we provide mechanistic insights regarding how Zfra overrides WWOX to induce cancer suppression and retard AD progression via Z cells.

## 1. Introduction to Molecular Signaling and Cell Fates

How a single cell maintains its highly complicated signaling network is essentially unknown. Signaling molecules are well connected in the interaction network that is essential in maintaining normal physiology. Alterations in signaling molecules, either in concentrations or structures, lead to disruption of the signaling pathway and the potential development of multiple diseases such as cancer, neurodegeneration, and others [1,2,3]. For example, an imbalance in L-tryptophan metabolism via the kynurenine pathway may affect and induce disorders ranging from cancer to neurodegenerative disease [1]. Dysregulation in the participating proteins in autophagy is linked to cancer development, neurodegenerative diseases, and immune defects [2]. NAD+ plays an indispensable role in signaling and metabolism. Controlled and balanced levels of NAD+ are needed to prevent metabolic diseases, cancer, neurodegeneration, and immune disorders [3].

Interestingly, inappropriate phase separation can cause aberrant formation of protein condensates or aggregates implicated in the pathogenesis of cancer and neurodegeneration [4,5]. Dysregulation of programmed cell death causes developmental disorders and may lead to cancer and neurodegeneration [6]. Nonetheless, there is still a long list showing molecule-regulated cancer and neurodegeneration. A straightforward question to ask is how those aberrant signaling pathways can travel down a similar path for the development of both cancer and neurodegeneration.

In this perspective review, we will go through the fundamental concept of chronic inflammation in neurodegenerative diseases and cancer, update the WWOX functional properties in suppressing Alzheimer’s disease (AD) and cancer, and discuss how Zfra peptide regulates AD and cancer via Hyal-2/WWOX/Smad4 signaling and how Z cell activation by Zfra works in limiting disease progression. 

## 2. Chronic Inflammation Is Constantly Associated with Neurodegeneration and Cancer 

Chronic inflammation is a common occurrence for the progression of neurodegeneration, as well as cancer growth [7,8,9,10,11]. Thus, suppression of inflammation may mitigate both diseases, but does not effectively provide a cure. Low-grade inflammation can be regarded as a process of aging. The aging event involves the production of inflammatory cytokines and activated immune cells. For example, the gut–brain axis is needed in the development of rotenone-induced Parkinson’s disease (PD) in rats, as revealed by changes in substantial intestinal histological alterations, such as shortened villi, crypt architecture loss, and inflammation, along with upregulation of inflammatory microglial cells [7]. Using techniques for determining microglial transcriptomes and epigenomes, one can estimate the altered microglial status affecting neuroinflammation, neurodegeneration, and disease progression [8]. In this case, specific alterations in AD and related diseases partly correlate with the status of inflammatory microglia [8]. Database analysis provides information regarding the effects of metformin on inhibiting oxidative stress, gluconeogenesis, and inflammation, linking to the mechanism for improving symptoms in AD and type 2 diabetes [9]. Most recently, the cGAS-STING pathway has been identified as a driving force for aging-related inflammation in peripheral organs and the brain [10]. Suppression of cGAS-STING signaling is likely to halt neurodegeneration in old age [10]. Again, suppression of inflammation in aging may not be able to stop the aging process, nor cancer growth, completely. 

Curcumin is good for preventing and suppressing neurodegeneration and cancer, as well as modulating immune functions [12,13]. Curcumin has tautomeric forms (Figure 1). The keto form of curcumin possesses antioxidant activity, whereas the enol form is not stable and tends to degrade. Curcumin induces apoptosis in cancer cells and reduces inflammation, angiogenesis, and tumor metastasis by targeting cancer signaling pathways such as p53, Ras, PI3K, AKT, Wnt-β catenin, and mTOR. The family of NADPH oxidase enzymes (Nox1-5, Duox1-2) has seven members, which are known to be involved in various diseases, such as inflammatory lung diseases, neurodegenerative diseases, and cancer. A novel Nox2 inhibitor, TG15-132, supports neuroprotection by blocking inflammatory responses, including the generation of reactive oxygen species and inflammatory cytokines [14]. Calebin-A is a natural polyphenol and a curcumin analog [15,16,17,18] (Figure 1). Calebin-A has a ferulic acid ester bond. Calebin-A supports neuronal survival from β-amyloid insult, inhibits cancer growth, and prevents obesity, partly due to its anti-inflammatory activities such as downregulating nuclear factor (NF)-κB activation [15,16,17,18]. Whether Calebin-A reduces peritumor coats in metastatic brain cancer or causes the degradation of amyloid plaques in the brain is unknown.

## 3. Patients Who Survive Cancer Do Not Develop AD

Epidemiological data show that AD and cancer could run in opposite directions. In most cases, patients who survive cancer do not develop AD. Conversely, patients with neurodegenerative diseases cannot develop cancer [19,20,21]. It has been estimated that AD patients have a 61% decreased risk of cancer incidents compared to reference subjects. It has been proposed that downregulation of Pin1 and Wnt signaling and upregulation of p53 contribute to neuronal death during cancer growth. By contrast, both Pin1 and Wnt are upregulated in cancer. This scenario fails to support the inverse relationship for AD and cancer development. 

Protein aggregation in the brain occurs frequently due to the downregulation of WWOX in the brain hippocampus in middle-aged individuals [22,23]. If the protein aggregation worsens with age, the individuals will likely have AD 30 to 40 years later due to the building up of β amyloid plaques and tau aggregates in the brain. If cancer develops simultaneously in middle age, cancer cells should have a faster formation of peritumor coats containing aggregated β amyloid and amyloid fibrils [24,25,26]. Hypothetically, the presence of amyloid fibrils in the peritumor coats in the newly developed tumors signals the brain to slow down the protein aggregation event. Metastatic cancer cells may relocate to the brain and cause neuronal death [25]. Under this condition, both cancer growth and neural death happen simultaneously. 

Drugs that are highly effective in curing AD and PD are not available. The rationale for repurposing anti-cancer drugs in treating AD, PD, and other neurodegenerative diseases is to speed up drug discovery for curing the diseases [27]. Many known kinase inhibitors have been proposed to be good for suppressing multiple protein kinases in neurodegenerative disorders. Nonetheless, whether these drugs can cure AD and PD remains questionable.

## 4. WWOX Exhibits Multiple Functional Properties 

WW domain-containing oxidoreductase (WWOX) was first isolated in 2000 [28,29,30], and has been considered a candidate tumor suppressor protein [28,29,30,31,32,33,34,35,36,37,38]. Since then, there have been many outstanding review articles related to this gene. The *WWOX/Wwox* gene and its encoded WWOX protein participate in cancer suppression [28,29,30,31,32] and inhibition of AD progression [22,23]. Whether WWOX protein controls the transition between cancer susceptibility and AD resistance, or cancer resistance and AD progression has not been entirely delineated.

### 4.1. WWOX Primary Structure and Binding Partners 

The wild-type WWOX protein possesses two *N*-terminal WW domains and a *C*-terminal short-chain alcohol dehydrogenase/reductase (SDR) domain [28,29,30]. A nuclear localization signal is located between the *N*-terminal first and second WW domains (WW1 and WW2). The WW domain is mainly responsible for protein/protein interactions. For example, WW1, possessing two tryptophan residues, binds proline-rich motifs PPXY or PPPY-containing target proteins for numerous biological functions during signaling [31,32,33,34,35,36,37,38,39,40,41]. Such events include cell death, differentiation, signaling, cell migration and recognition, and disease development [10,11,12,13,14,15,16,17,18,19,20,21,22]. WW2 has one tryptophan, whose function has not been elucidated. However, WW2 may team up with WW1 to maintain an appropriate tertiary conformation that affects the function of WW1 in protein/protein binding [42]. Nuclear localization of WWOX may occur, especially when cells are stimulated with growth factors or are under apoptotic stress [35]. The crystal structure of WWOX has not been established.

### 4.2. WWOX Is Not a Typical Tumor Suppressor

The human *WWOX* gene, encoding the WWOX protein, is located on a common fragile site, i.e., *FRA16D*—the second most frequent site of its kind [28,29,30,31,32]. Most cancer cells derived from breast, lung, prostate, and other organs have alterations in the *WWOX* gene [32,33,34,35,36,37,38,39]. Restoration of the *WWOX* gene and the resulting protein in cancer cells blocks their growth in vivo and in vitro [40,41]. WWOX protein is not a typical tumor suppressor, as it participates in numerous biological events, including (i) cell survival, proliferation, differentiation, cell cycle regulation, and senescence via complicated signaling pathways [38,39,40,41,42,43,44,45], (ii) aging and neurodegeneration [22,23,46,47], (iii) apoptotic cell death [30,48,49,50,51], (iv) chromosomal DNA stability [52,53], (v) bubbling cell death [54,55,56], and (vi) cell-to-cell recognition and migration [57]. Human newborns lacking the *WWOX* gene and functional WWOX protein suffer severe neural diseases but do not have spontaneous tumor formation, suggesting *WWOX* does not fit Knudson’s two-hit hypothesis of tumorigenesis [31,35,45,46]. 

### 4.3. pY33-WWOX Maintains Normal Mitochondrial Physiology

Under physiological conditions, a portion of activated Tyr33-phosphorylated WWOX (pY33-WWOX) localizes on the outer membrane of the mitochondria to support normal physiology [28,48,58,59,60,61]. Without WWOX, cells undergo uncontrollable proliferation and die readily [43]. Under stress conditions such as exposure to UV irradiation and chemotherapeutic drugs, significantly upregulated pY33-WWOX, together with p53, induces mitochondrial apoptosis [58,59,60,61] or co-relocates to the nucleus to exert cell death [30,35,47,48]. The prosurvival TNF receptor-associated factor 2 (TRAF2) blocks the apoptotic function of WWOX and p53 [54]. The trafficking protein particle complex 6A (TRAPPC6A or TPC6A) carries WWOX to undergo nuclear translocation [22]. The protein complex dissociates in the nuclei. WWOX remains in the nuclei, and TPC6A continues to relocate to the nucleoli. Indeed, during trafficking, TPC6A travels from the mitochondria to the nucleoli and then from the nucleoli back to the mitochondria [22]. Carrying of TPC6A with proteins and associated materials between mitochondria and nucleoli is unusual but is essential for normal cell physiology [22]. The nucleolus is a center of ribosome synthesis and is abundant in small nucleolar RNA for pre-rRNA processing.

### 4.4. Overexpressed pY33-WWOX Induces Apoptosis Which Is Unfavorable for Neurons But Beneficial for Eliminating Cancer Cells In Vivo

The binding of pY33-WWOX with target proteins does not require the presence of PPXY or PPPY motif in the target proteins. Indeed, pY33-WWOX expands its scope in protein binding interactions [32]. In vivo data confirms that pY33-WWOX binds and functions together with many nuclear transcription factors such as p53 [30,48,49], CREB 1 (CAMP responsive element binding protein 1) [50], and c-JUN [50], and JNK1 (c-JUN N-terminal kinase 1) [48,50]. These connections may be linked to neuronal death. For example, during the acute phase of sciatic nerve dissection, the activation of JNK1 and WWOX occurs quickly, which leads to rapid apoptosis of large neurons in the ipsilateral side of the injury in rat brains [50]. In the chronic phase of injury, small neurons in both the ipsilateral and contralateral sides of dorsal root ganglia (DRG) have co-activation of WWOX, CREB, and NF-κB, which leads to apoptosis [50] (Figure 2).

### 4.5. WWOX Modulates the Activities of Transcription Factors to Determine Neuronal Survival or Death 

WWOX controls chromosomal DNA stability [52,53]. During sciatic nerve dissection, activated pY33-WWOX is co-localized in the nuclei with several transcription factors: CREB, c-JUN, ELK-1, CRE, and AP-1 [50,62]. It is very likely that these proteins co-relocate to the nuclei simultaneously. For example, at 6 hr post-sciatic nerve dissection, accumulation of pY33-WWOX goes up 40 to 65% in the nuclei of medium/large neurons in the ipsilateral DRG for at least two months [50]. p-JNK1 takes 24 h to reach a similar extent of nuclear accumulation in the medium/large neurons, followed by a reduction on day 7. c-JUN accumulation in the nuclei of medium/large neurons also takes 6 h and lasts one week only. Interestingly, ATF3 accumulation in the nuclei of medium/large neurons by more than 65% starts in 24 h and lasts two months. Transcription factors p-CREB, NF-kB, and SMAD4 have little or no accumulation in the medium/large neurons in the ipsilateral dorsal root ganglia. Functional suppression of p-JNK1 by pY33-WWOX rapidly causes apoptosis of the medium/large neurons during the acute phase of nerve injury [48,50,63].

In vitro promoter assay revealed that transiently overexpressed WWOX activates the promoter activity driven by transcription factors c-JUN and Elk-1 [50] (Figure 2). Similarly, WW1 activates the promoter of NF-κB [50]. Again, these interactions cause neuronal apoptosis. In contrast, WWOX inhibits the activation of promoter elements induced by prosurvival transcription factors CREB, CRE, and AP-1, leading to neuronal death [50]. 

Retarded death of small neurons in two months at the chronic phase of sciatic nerve dissection is probably due to accumulation and functional balance of pY33-WWOX, ATF3, and p-CREB in the nuclei [50] (Figure 2). ATF3 is either protective or apoptosis-inducing for neurons. A nuclear calcium-CREB signaling to an ATF3-mediated neuroprotective gene repression program counteracts age- and disease-related neuronal loss [64]. Nonetheless, ATF3 modulates inflammatory responses during injury, leading to cell death [65,66,67].

Overall, pY33-WWOX orchestrates an array of transcription factors during sciatic nerve dissection. Binding of activated pY33-WWOX with pro-survival JNK1 [48,63], NF-κB [50], CREB [50], and AP-1 results in inhibition of their protective function, thereby leading to neuronal death (Figure 1). Meanwhile, pY33-WWOX enhances the promoter activation driven by c-JUN and ELK-1 to induce neuronal apoptosis. pY33-WWOX physically binds c-JUN [50], whereas its binding with ELK-1 is unknown. We do not exclude the possibility of the team effort of pY33-WWOX and pS46-p53 in manipulating the function of transcription factors. pY33-WWOX binds pS46-p53, and both proteins induce apoptosis in a synergistic manner [49].

## 5. Zfra4-10 Activates Hyal-2/WWOX/Smad4 Signaling Pathway for Retardation of Neurodegeneration

We reported that when endogenous WWOX strongly binds intracellular protein partners in cancer cells, the cells cannot grow effectively in vivo [68]. When mice or cells are stimulated with the Zfra4-10 peptide, this peptide binds cell membrane hyaluronidase type II (Hyal-2) to activate the Hyal-2/WWOX/Smad4 signaling pathway [56,69,70,71,72,73,74]. The signaling activation leads to Smad4-mediated cell death or survival [69,70,71,72,73,74], cancer suppression [70,71,73,74], and restoration of memory loss in AD [71], depending upon the strength of signal activation. 

### 5.1. Zfra Induces Activation of Spleen Hyal-2+ CD3- CD19- Z Cell

The Hyal-2/WWOX/Smad4 signaling induces spleen Hyal-2+ CD3- CD19- Z cell activation for eradicating cancer cells [70]. While Z cell function in vivo is largely unknown, our laboratory has been focusing on characterizing the role of Z cells in suppressing neurodegeneration in AD, PD, and seizures. Z cells are so named because Zfra activates this cell type in the spleen [70,71]. Z cells were originally isolated from T/B cell-deficient NOD-SCID mice. Z cells are not T, B, monocyte, or NK cells, but are similar in size to T and B cells. We have determined the gene expression profile of naïve Z cells (see GEO database Accession: GSE98409, ID: 200098409). Z cells are a new type of lymphoid cells. Activated Z cells enable immune-deficient NOD-SCID mice to suppress cancer growth [70,71]. 

Zfra-activated Z cells cause cancer cell death with or without physical contact [70]. When cancer cells are co-cultured with activated Z cells in vitro, Z cells undergo clonal expansion and rapidly eradicate cancer cells [68]. Without prior exposure to cancer cells or cancer cell antigens, Zfra-activated Z cells effectively induce cancer cell death [70]. Non-activated Z cells cannot kill cancer cells even though they relocate, together with T cells, to the cancer lesions [68]. Both T and Z cells group side-by-side in cancer lesions [68]. 

Unlike nature killer (NK) cells, naïve Z cells require pre-activation by Zfra1-31 or 4-10 or WWOX7-21. This activation allows Z cells to recognize more than ten cancer cell lines for effective eradication [70]. Thus far, Zfra1-31 or 4-10, and WWOX7-21 peptides, and antibodies against Hyal-2 or pY216-Hyal-2 are known ligands for activating Z cells [68]. Inflammatory NK cells kill cancer cells and cause damage to neurons [75,76,77,78]. In contrast, activated Z cells protect neurons from being damaged [71]. The activation markers for Z cells are being determined.

### 5.2. Zfra Restores Memory Loss in Triple Transgenic Mice for AD 

Zfra4-10-mediated restoration of memory loss in triple transgenic mice for AD (3xTg) occurs via its inhibition of the aggregation of amyloid beta 42 (Aβ42), accelerating the degradation of aggregated proteins and blocking the activation of inflammatory NF-κB [71] (Figure 3). Zfra4-10 prevents the age-dependent progression of AD-like symptoms in 3xTg mice [72]. Whether Zfra4-10 suppresses the expansion of inflammatory glial cells in the brain is unknown. In addition to amyloid beta, there are numerous cytosolic proteins subject to degradation upon binding with Zfra—an action designated as zfration [68,70,71,72,73,74]. 

While the action of pS14-WWOX is still being determined, it is not appropriate to consider pS14-WWOX a disabled form. pS14-WWOX is needed for T/B cell differentiation [45]. However, the pS14-WWOX-induced T/B cells cannot target cancer cells for eradication.

Full-length Zfra1-31 and the truncated Zfra4-10 undergo polymerization when the peptides are suspended in phosphate-buffered saline (PBS) but not in water [70,71]. The presence of Cys9 and Cys12 in the Zfra peptide suggests covalent binding occurs in both intra- and inter-molecular manners [70,73] (Figure 4A). Excessive polymerization of Zfra makes the peptides ineffective in blocking cancer growth [70,73]. Whether overly polymerized Zfra blocks neurodegeneration is unknown. Polymerized Zfra cannot be separated into monomers by reducing chemicals such as β-mercaptoethanol, suggesting the presence of another type of covalent binding. The likely candidate is the phosphorylation site Ser8. Compared to non-phosphorylated Zfra, pS8-Zfra is not functionally active. 

Our research on Zfra peptides in mice has revealed a fascinating phenomenon. When these peptides are administered, they quickly bind with plasma proteins, forming complexes that are then sequestered in the spleen. This leads to the activation of spleen Hyal-2+ CD3- CD19- Z cells in vivo [70], a process that can last for several months due to the resistance of the Zfra/protein complex to degradation. These activated Z cells may then migrate to various organs, including the liver and lungs. However, what intrigues us is the potential presence of Z cells in the small intestine. If they are indeed found in the gut, their activation could play a role in halting neurodegeneration via the gut–brain axis. 

Additionally, activated Z cells do exist in the brain [71]. Whether these brain-activated Z cells come from the spleen is unknown. The role of brain Z cells in preventing neurodegeneration is of great interest to investigate.

### 5.3. Zfra Physically Interacts with Endogenous Proteins for Degradation 

Zfration is critical for protein degradation [70,71,72,73,74]. Full-length Zfra1-31 proteins bind WWOX at the *N*-terminal WW1 domain and the *C*-terminal SDR domain [73], which was established by yeast two-hybrid analysis. The binding of Zfra1-31 with the recombinant WW1 domain is covalent [68] (Figure 4B). Whether the Zfra/SDR complex is covalent needs investigation. When cell lines were used to analyze Zfra-mediated protein degradation, Zfra peptides were added to the cell lysates. Covalent Zfra/WWOX or protein complexes can be seen by observing banding patterns with reducing and non-reducing gels. Compared to control cell lysates, Zfra enhances protein degradation with time. Known proteinase inhibitors and a proteasome inhibitor MG-132 cannot block the protein degradation, suggesting that a novel protease degrades the zfrated proteins.

### 5.4. Zfra4-10 Covalently Interacts with WWOX7-21

A portion of cytosolic WWOX is localized on the cell membrane, and WWOX7-21 is a surface-exposed epitope [74]. Synthetic Zfra4-10 and WWOX7-21 peptides powerfully suppress the growth and metastasis of 4T1 breast cancer cells in mice [68]. However, when combined, both Zfra4-10 and WWOX7-21 peptides lose their function in blocking 4T1 growth in mice, suggesting that both peptides functionally antagonize each other via covalent binding [68]. 

### 5.5. Zfra4-10 and WWOX7-21 Strengthen the Binding of Endogenous WWOX with Target Proteins in Stem Cells, Microglia Cells, Astrocytes, and Treg Cells and in Exosomes In Vivo

We reported that the more strongly endogenous WWOX binds intracellular protein partners, the more weakly the cancer cells can grow in vivo [68]. When mice receive either Zfra4-10 or WWOX7-21 peptide via tail vein injections, endogenous WWOX strongly increases its binding (1- to 7-fold increases) with C1qBP, CD133, p21, JNK1, COX2, p-ERK, Foxp3, and p53 in the spleen cells, as determined by co-immunoprecipitation using organ lysates (Figure 5A,B). The binding correlates with suppression of cancer growth [68] (Figure 5A–C). The stronger the pY33-WWOX binding of partner proteins, the better the suppression of cancer growth in vivo [68] (Figure 5D). In stark contrast, when mice receive both Zfra4-10 and WWOX7-21 peptides, a dramatically reduced binding of WWOX with the target proteins occurs, down to the baseline level or none. In the lung, Zfra4-10 or WWOX7-21 peptide alone also strongly enhances the binding of endogenous WWOX with Iba1, Oct4, ERK1/2, NF-κB p65, GFAP, and p53 (2- to 5-fold increases).

Upregulation of CD133 or OCT4 expression indicates expanding stem cell populations in the spleen. Foxp3-positive T regulatory (Treg) cells are also expanded in the spleen. Interestingly, Iba1-positive microglia cells and GFAP-positive astrocytes are present in the lungs. Whether these cells can relocate to the brain is unknown.

While we used organ lysates for co-immunoprecipitation, we will repeat the experiments using tissue sections and examine the binding via antibody FRET imaging or affinity proximity assay. This will detail not only the binding affinity, but also the extent of prevalence in binding among cell types.

### 5.6. Inhibition of S14 Phosphorylation in WWOX by Zfra4-10 or WWOX7-21 Leads to Reduced Neurodegeneration

Overexpressed pY33-WWOX is proapoptotic, whereas an optimal level of pY33-WWOX is essential for maintaining normal physiology [48,49]. When the level of pY33-WWOX goes down, the pS14-WWOX level goes up in vivo [70,71,72]. pS14-WWOX supports the progression of cancer development, growth, and metastasis [70] and enhances the progression of AD [71,72]. pS14-WWOX is accumulated in the lesions of cancers and AD brains. Zfra4-10 strongly reduces the level of pS14-WWOX and thereby abolishes cancer growth and retardation of AD progression [70,71,72]. WWOX7-21 is also potent in blocking WWOX phosphorylation at S14. Conceivably, both Zfra4-10 and WWOX7-21 induce or activate a specific phosphatase(s) that causes pS14 dephosphorylation. The specific phosphatase(s) has yet to be identified. 

While the action of pS14-WWOX is still being determined, it is not appropriate to consider pS14-WWOX as a disabled form. pS14-WWOX is needed for T/B cell differentiation [45]. However, induced T/B cannot block cancer growth.

## 6. Dramatic Upregulation of pY33- and pY287-WWOX in the Brain Cortex of Heterozygous *Wwox* Mice

Thus far, no reports have demonstrated that WWOX becomes aggregated in the brain. Compared to wild-type mice, heterozygous *Wwox* mice express half as much WWOX protein in tissues and organs. Most interestingly, WWOX undergoes enhanced phosphorylation at Y33 and Y287 in the brain cortex of heterozygous *Wwox* mice (~1-fold increase), but not in the brains of wild-type mice [72]. The increased phosphorylation of Y33 and Y287 in WWOX in the heterozygous mice is necessary to maintain normal physiological functions. In general, heterozygous *Wwox* mice normally behave like wild-type mice, although these mice have an accelerated neurodegeneration [71]. No increase in S14 phosphorylation of WWOX is shown [78], suggesting an inter- or intra-molecular autoregulatory mechanism for WWOX functional activation and turnover [57]. That is, binding interactions can be found for the WW or SDR domains by itself or WW/SDR self-folding or intermolecular interactions [57]. Both pY33- and pY287-WWOX do not exhibit aggregation in the brain cortex. ACK-1 phosphorylates WWOX at Y287, and pY287-WWOX can be ubiquitinated for proteasomal degradation [79]. Another report showed that when cells are under DNA single-strand break checkpoint activation, WWOX is ubiquitinated at K274 by the ubiquitin E3 ligase ITCH and interacts with ataxia telangiectasia-mutated (ATM) [53]. WWOX can undergo modification by small ubiquitin-like modifier (SUMO) proteins. As a result, the SUMOylated WWOX blocks prostate cancer growth [80]. 

### 6.1. pT12-WWOX Is an Aggregated form in the AD Brain Lesions

We have determined for the first time that pT12-WWOX aggregates or plaques are found in the neocortices and cortices of 11-month-old heterozygous *Wwox* mice but not in those of wild-type mice [72] (Figure 6A). The pT12-WWOX aggregates are about 30 to 60 μm in diameter, which are formed most likely at an earlier age in mice or at middle age in humans. Presumably, pT12-WWOX aggregates cause the activation of the protein aggregation cascade starting from the polymerization of TRAPPC6AΔ (or TPC6AΔ), followed by TIAF1 (TGFβ-induced antiapoptotic protein), and SH3GLB2 (SH3 domain-containing GRB2 Like, endophilin B). The resulting complexes lead to caspase activation, degradation of amyloid precursor protein (APP), generation of amyloid beta, and neuronal apoptosis [22,81,82] (Figure 6B). Compared to the wild-type, TPC6AΔ isoform has an internal deletion of 14 amino acids in the *N*-terminus [45].

### 6.2. pT12- and pS14-WWOX in Controlling Protein Aggregation?

The protein aggregation cascade may start in middle age and keep going for 20 to 30 years before the appearance of Alzheimer’s disease symptoms [22,81,82]. pS14-WWOX is involved in disease progression [31,61]. While T12 and S14 are close to each other, their relationship regarding phosphorylation and conformational changes of WWOX is unclear. However, both amino acid residues are involved in AD progression. In principle, pY33-WWOX blocks the aggregation of TPC6AΔ and related downstream proteins [22,81,82]. pT12-WWOX is likely to stimulate the aggregation cascade of TPC6AΔ, TIAF1, and SH3GLB2. When neuroblastoma SK-N-SH cells are treated with neurotoxin 1-methyl-4-phenylpyridinium (MPP+) for inducing PD-like symptoms, TPC6AΔ expression is increased and becomes polymerized, which causes aggregation of TIAF1, SH3GLB2, Aβ, and tau [72]. Similarly, transforming growth factor beta (TGF-β) stimulates the polymerization of TPC6AΔ, TIAF1, and SH3GLB2 in the aggregation cascade. Notably, Zfra4-10 peptide blocks the MPP+ effects and thereby sustains the survival of neurons. 

Taken together, when WWOX protein is downregulated in the brain cortex, TPC6AΔ is expected to have an accelerated aggregation as shown in the *Wwox* knockout mice. The aggregated TPC6AΔ is at the crossroad for leading to the progression of AD or PD [72]. How TPC6AΔ directs the signaling for going to AD or PD remains to be determined. Alternatively, upregulation of pT12- or pS14-WWOX and downregulation of pY33-WWOX may compete for turning on or off to the road for AD or PD.

## 7. WWOX in Embryonic Development and AD Progression

WWOX plays a crucial role in affecting embryonic development. Functional deficiency or defects in the *WWOX* gene and protein lead to severe neural disorders, metabolic disorders, mental retardation, neurodegeneration, immune defects, stunted growth, and early death [46,47,72,83,84]. *WWOX* gene deficiency from bi-allelic alterations in chromosome 16q leads to autosomal recessive spinocerebellar ataxia 12 (SCAR12) and WWOX-related epileptic encephalopathy (WOREE) syndromes [46,47,72,83]. Children suffering from SCAR12 have early childhood onset of cerebellar ataxia and difficulty in coordinating the movement of their muscles. WOREE is caused by bi-allelic gene mutations. Newborn patients suffer intractable epileptic seizures, severe defective development in the neural system, and movement disorders. 

The *WWOX* gene is also regarded as a risk factor for AD [85] and is associated with the progression of PD [86]. In AD, downregulation of WWOX occurs in the hippocampus in middle age, leading to very slow and gradual aggregation of a protein cascade group over 20 to 30 years [22,78,81,82]. Balanced phosphorylation among T12, S14, Y33, and probably Y34 in the WW1 is critical to limiting disease manifestation [70,71]. For example, when Y33 phosphorylation in WWOX is downregulated, S14 phosphorylation is then upregulated. Aggregation of pT12-WWOX is likely blocked by pS14- or pY33-WWOX. Conceivably, pY34 inhibits the action of pY33. pS14-WWOX promotes the progression of cancer and AD [70,71]. Nonetheless, these presumed functions remain to be established. 

Furthermore, tau hyperphosphorylating enzymes JNK and ERK can be bound and functionally blocked by Y33-phosphorylated WW1. WWOX prevents tau hyperphosphorylation by these enzymes [23]. The SDR domain binds GSK3β and the C-terminal region of tau [23,87]. This binding leads to inhibition of tau hyperphosphorylation [23,87].

## 8. WWOX Antagonizes p53 for Inducing Neurodegeneration In Vivo

Protein functional antagonism may lead to the development of neurodegeneration. For example, both p53 and WWOX may functionally antagonize in regulating cancer growth, and the antagonism can lead to neurodegeneration in vivo [88]. To achieve the effect of neurodegeneration, p53 and/or WWOX cDNA expression constructs are overexpressed in p53-deficient lung cancer cells and are then inoculated in mice. Ectopic WWOX strongly inhibits lung cancer growth and inflammatory reaction in vivo [88]. In contrast, p53 does not have an inhibitory effect. In combination, the inhibitory function of WWOX is nullified by p53. Most strikingly, when mice receive p53/WWOX-expressing lung cancer cells, these mice are shown to have BACE (β-secretase 1) upregulation, APP degradation, tau tangle formation, and amyloid β generation in the brain and lung [88]. Hence, the functional opposition between p53 and WWOX leads to enhanced cancer growth and accelerated neurodegeneration in vivo [31,46,88]. Under stress conditions, pS46-p53 physically interacts with pY33-WWOX [49]. This strongly associated complex appears to be mainly responsible for inducing apoptosis [49]. 

However, when pY33-WWOX is de-phosphorylated and S14 phosphorylated, the pS46-p53/pS14-WWOX complex could have an opposite effect. Thus, upregulation of pS14-WWOX facilitates the growth of cancer and the progression of AD [70,71]. The specific kinase and phosphatase involved in pY33-WWOX dephosphorylation and pS14-WWOX phosphorylation is unknown. Finally, Zfra4-10 or WWOX7-21 downregulates pS14-WWOX [68,70,71], suggesting that either peptide may covalently bind pS46-p53/pS14-WWOX complex for proteolytic degradation.

## 9. Membrane Epitopes WWOX7-21 and WWOX286-299 and Functional Implications 

WWOX is located ubiquitously in intracellular locations. Computational prediction does not show that WWOX protein has a membrane localization signal sequence. Membrane Hyal-2 binds the WW1 domain and anchor WWOX to the membrane area [56,69]. In addition, WWOX is anchored by Ezrin to the cytoskeletal-membrane area [89]. WWOX can undergo self-polymerization in the cell membrane, as confirmed by immunoelectron microscopy [56,69]. The self-polymerization is due to binding of WW1 with WW1, SDR with SDR, or WW1 with SDR in an intermolecular or intramolecular manner [57].

### 9.1. WWOX7-21 Epitope Confers Cancer Suppression and Probably Blocks AD Progression

Structurally, there are two epitopes in the membrane-localized WWOX, which are at amino acid 7 to 21 and 286 to 299. Synthetic peptides WWOX7-21 and WWOX286-299 and corresponding antibodies were used in functional characterization [57,90]. Treatment of 4T1 breast cancer cells with antibody against the WWOX7-21 epitope makes the cells susceptible to ceritinib, UV irradiation, and many chemotherapeutic drugs [73]. In contrast, WWOX7-21 peptide significantly increases the death of cultured breast 4T1 cells and cell sphere explosion mediated by ceritinib [73]. This peptide, which is as short as WWOX7-11, has even stronger potency in suppression and prevention of the growth and metastasis of melanoma and skin cancer cells in mice [73]. Both peptides probably mediate the signaling from membrane Hyal-2/WWOX and then recruit Smad4 for blocking cancer growth. WWOX7-21 peptide can self-polymerize [57,90], suggesting that the exogenous WWOX7-21 peptide binds the membrane WWOX7-21 epitope and probably Hyal-2 for initiating the signaling. Hyal-2 physically binds the *N*-terminal first WW domain in a Y33 phosphorylation dependent manner [56]. Alteration of Y33 in WWOX probably abolishes its binding with Hyal-2.

Additionally, WWOX7-21 peptide reduces ERK phosphorylation, upregulates proapoptotic pY33-WWOX, induces calcium ion influx, and abolishes IkBα/WWOX/ERK prosurvival signaling [90]. Consequently, these events lead to cancer suppression. While pS14-WWOX supports cancer growth and enhances AD progression, pS14-WWOX7-21 peptide significantly enhances cancer growth in vivo and blocks ceritinib-mediated apoptosis in vitro [73]. In parallel, antibody against pS14-WWOX7-21 peptide reduces cancer growth [73]. Zfra peptide is potent in inducing neuroprotection [71]. Like Zfra peptide, WWOX7-21 peptide activates the Hyal-2/WWOX/Smad4 signaling, suggesting that WWOX7-21 can exert neuroprotection.

### 9.2. WWOX7-21 and WWOX286-299, along with Membrane Type II TGFβ Receptor (TβRII), Control Cell Migration and Cell-Cell Recognition

Cells expressing functional WWOX (WWOXf) migrate collectively and can fend off the individually migrating WWOX-deficient or -dysfunctional (WWOXd) cells to undergo retrograde migration [57,90]. Interestingly, the retrograde-migrating WWOXd cells kill a portion of WWOXf cells from a remote distance, without physical contact. Mechanistically, WWOXd cells dramatically increase the redox activity in WWOXf cells, thereby causing cell death [90]. WWOXd cells utilize the IκBα/ERK/WWOX signaling to survive [57,90]. 

Specifically, when cells have surface exposure to the WWOX7-21 epitope, these cells attract any approaching WWOXf or WWOXd cells of the same or different species [57]. We have identified MIF (macrophage migration inhibitory factor) as an extracellular sensing molecule [90]. When cells are exposed to WWOX286-299 epitope on their surface, these cells strongly fend off WWOXd cells [57]. Membrane WWOX also binds membrane type II TGFβ receptor (TβRII). Stimulating WWOXf cells with TβRII IgG antibodies leads treated cells to greet WWOXd cells to merge with each other. However, when WWOXd are pretreated with TβRII IgG antibody, these cells lose recognition by WWOXf cells. WWOXf cells can kill the TβRII antibody-treated WWOXd cells. The observations suggest that normal cells can be activated to attack metastatic cancer cells. Together, membrane WWOX/TβRII complex is needed for cell-to-cell recognition, maintaining the efficacy of Ca^2+^ influx and control of cell invasiveness.

### 9.3. WWOX in Cortical Neuron Migration and Loss of WWOX Causes Neuronal Heterotopia

Whether WWOX participates in deciding neuronal migration and positioning in the brain cortex during embryonic development is largely unknown. However, the fact is that newborn patients with WWOX deficiency develop neuronal heterotopia and severe epileptic seizures [83,84,91,92,93]. Without WWOX protein, neurons accelerate their migration and accumulate in the neocortex, known as neuronal heterotopia [84,94,95]. Moreover, reduced GABA-ergic inhibitory interneurons has been demonstrated in *Wwox* knockout mice [96]. Many genes in common fragile sites are unstable and tend to undergo deletions and alterations [97]. WWOX, DAB1, and many proteins are in involved in neuronal migration and lamination in the developing cerebral cortex [97,98,99,100]. 

Here, we propose that during cortical lamination, WWOX controls neuronal layer development via transition between epitope exposure. For example, surface exposure of the WWOX7-21 epitope in a group of cells allows the other cell group to migrate forward, and the WWOX286-299 epitope signals the other cells to stop moving. This scenario is based upon the aforementioned findings [57,90]. Nonetheless, the relationship can be complicated by phosphorylation in the epitopes such as pT12, pS14, and pY287. Given the role of Zfra in suppressing WWOX phosphorylation at T12, S14 and Y287 and inhibiting the formation of neuronal heterotopia [84], Zfra is promising in restoring neuronal functions.

## 10. Z Cells Have Memory Function in Eliminating Cancer Cells

### 10.1. Z Cell Activation by Zfra4-10, WWOX7-21, and Sonicated Hyaluronan HAson8 to Kill Cancer Cells and Retard AD Progression

Spleen Z cell activation is critical in killing cancer cells. Z cells can be activated by Zfra4-10, Zfra1-31, and WWOX7-21 peptides [68,73]. In addition, 8-h-sonicated hyaluronan (HAson8) and antibody against Hyal-2 or pY216-Hyal-2 activate Z cells to kill cancer cells [68]. Native hyaluronan has no effect [68]. These agonists bind membrane Hyal-2 and then activate the Hyal-2/WWOX/Smad4 signaling in Z cells [68]. Z cell activation requires dephosphorylation of WWOX at S14, Y33 and Y61 [45,61]. Activated Z cells exhibit memory function. How the agonists establish anti-cancer memory function in Z cells remains to be determined, because Z cells do not require pre-exposure to cancer antigens. How Z cells acquire cytotoxic capability in eradicating cancer cells in vivo and in vitro is also yet to be revealed [68]. By the same token, we strongly believe that activated Z cells have the ability to protect neurons from being damaged or degenerated.

### 10.2. Zfra4-10 Activates Z Cells in Immune Deficient NOD-SCID Mice

Zfra activates Z cells in normal mice and T/B cell-deficient NOD-SCID and T-deficient nude mice [68,70,71]. Zfra-activated Z cells relocates to cancer lesion to eliminate cancer cells in both immune efficient and deficient mice [68,70,71]. That is, activated Z cells restore the immune response in immunodeficient mice to limit cancer growth and probably retard age-dependent neurodegeneration. 

A population of Z cells are located in the brain. This cell population can be isolated by flow cytometry. Whether these cells come from other organs or are the residential cells in the brain is unknown. Once transferred to tumor-growing mice, the purified brain-activated Z cells cause cancer cell death. Their role in mitigating the AD-like symptoms in mice is being investigated. Importantly, we are now able to activate the Z cell immune response in so-called immune-deficient mice for protection against neurodegeneration, cancer, and probably diabetes.

### 10.3. Zfra1-31 for Treating Hyperglycemia/Diabetes-Associated Neurodegeneration

A recent report determined that Zfra1-31 potently treats hyperglycemia/diabetes-associated neurodegeneration [101]. Diabetic rats at six months old have high levels of activated WWOX in the brain cortex and hippocampus compared to normal rats. The association of WWOX with the regulation of glucose metabolism has been reported [102]. Indeed, the WWOX/HIF1A axis is involved in altering glucose metabolism [102,103,104,105,106,107]. Substantial evidence from a genome-wide association study (GWAS) revealed the critical association of WWOX with diabetes and glucose metabolism [108,109,110,111,112]. For example, when the WWOX/HIF1A axis is downregulated, glucose metabolism is altered, and the changes may allow the development of metabolic disorders. Also, in early diabetic retinopathy, the knockdown of *WWOX* by specific siRNA in vitro suppresses superoxide production caused by the high glucose in photoreceptors from diabetic mice for two months [112]. Ablation of the *WWOX* gene in skeletal muscle leads to altered glucose metabolism [106].

WWOX becomes Y33-phosphorylated in response to high glucose, which leads to mitochondrial apoptosis. Indeed, overly expressed pY33-WWOX can induce apoptosis in vivo [50,51,58]. Based upon our observations [68,70,71], Zfra blocks the phosphorylation and covalently binds WWOX for degradation [46,72]. As a result, activated WWOX-mediated mitochondrial apoptosis is blocked. Overall, this finding is important and yet not surprising for Zfra. Zfra can be a universal drug for targeting many diseases via Hyal-2/WWOX/Smad4 signaling.

## 11. Summary and Concluding Remarks

We explored the possibility of Zfra in restoring cell function under WWOX functional deficiency or absent expression. We determined that Zfra overrides WWOX in suppressing cancer growth and the progression of neurodegeneration. First, a key mechanism is that Zfra4-10, WWOX7-21, and HAson8 induce the memory anti-cancer Z cells [24,25,26,27,28] (Route **i**) (Figure 7). Whether other types of immune cells are involved remains to be identified. Second, Zfra4-10 and WWOX7-21 induce the intracellular formation of the WWOX complex with target proteins [68] (Route **ii**). The stronger the binding of WWOX with target proteins, the better the suppression of cancer growth and retardation of neurodegeneration [68]. The turnover rate of WWOX-regulated protein complexes is unknown and remains to be elucidated. Finally, Zfra rapidly and covalently conjugates with target proteins by zfration [70,71], and the resulting complexes undergo degradation independently of ubiquitination and proteasomal degradation (Route **iii**). It is unclear whether the WWOX-conjugated protein complexes are stable and resist zfration for further degradation or the complexes are subjected to ubiquitination/proteasomal degradation. These scenarios remain to be determined.

### 11.1. pT12-WWOX as an Initiator of Protein Aggregation Cascade?

pS14-WWOX supports the progression of many diseases such as cancer and Alzheimer’s disease [70,71]. pS14-WWOX does not appear to undergo aggregation. A recent intriguing discovery is that pT12-WWOX occurs as aggregates in the cortex of heterozygous *Wwox* mice, but not in wild-type mice [78]. How pT12-WWOX becomes aggregated is unknown. pT12-WWOX aggregates could be in the middle of an aggregation cascade. Alternatively, as an initiator, phosphorylated-T12 in the first WW domain area undergoes non-stop self-polymerization. 

Whether the co-presence of pS14, pY33, pY34, and pY287 with pT12 can occur in the same WWOX molecule is unknown. Perhaps co-phosphorylation among all the phosphorylation sites in WWOX is necessary to stabilize the protein and prevent self-aggregation. The WW1 domain has antiparallel β-sheets, which can undergo self-binding [57]. The crucial conditions, such as changes in intracellular pH, ion concentrations, and abnormal WWOX-binding proteins, may lead WWOX to extensively self-polymerize via the first WW domain and aggregate eventually. 

While T12 and S14 are close to each other, we suspect pT12 may nullify the pathogenic function of pS14 in WWOX. Under this case, pT12 reduces the severity of pS14 in supporting disease progression such as AD development [5,53,56]. What remains to be established are the scenario and molecular mechanisms for the induction of TPC6AΔ polymerization by pT12-WWOX aggregates and the degradation of pT12-WWOX aggregates by Zfra.

Another intriguing finding is that pY33-WWOX can be released from the neurons to the extracellular matrix in the cortex [72]. pY33-WWOX is critical in maintaining cellular functions and stabilizing the normal physiology of mitochondria with p53 [9]. A portion of cytosolic pY33-WWOX is localized to the cell membrane [57]. Release of this protein is likely. We will examine whether the released pY33-WWOX binds amyloid plaques in the extracellular matrix.

### 11.2. Concluding Remarks

Synthetic Zfra peptides, Zfra1-31 or Zfra4-10, effectively prevent and block cancer growth in vivo. Zfra4-10 strongly retards the progression of AD-like symptoms in 3xTg mice during aging [71]. For treatment, Zfra4-10 peptide restores memory loss in 9-month-old 3xTg mice by blocking the aggregation of a group of initiating proteins, which aggregate like a cascade. Zfra activates Hyal-2+ CD3- CD19- Z cell to cause cancer cell death. Z cells are downstream effectors of Zfra-mediated inhibition of neurodegeneration and have great potential for cell therapy in AD or other neurodegenerative diseases. *WWOX*-deficient newborns suffer severe neurodegeneration. Zfra may override *WWOX* deficiency in restoring normal physiological functions.

## Figures and Tables

**Figure 1 ijms-25-03507-f001:**
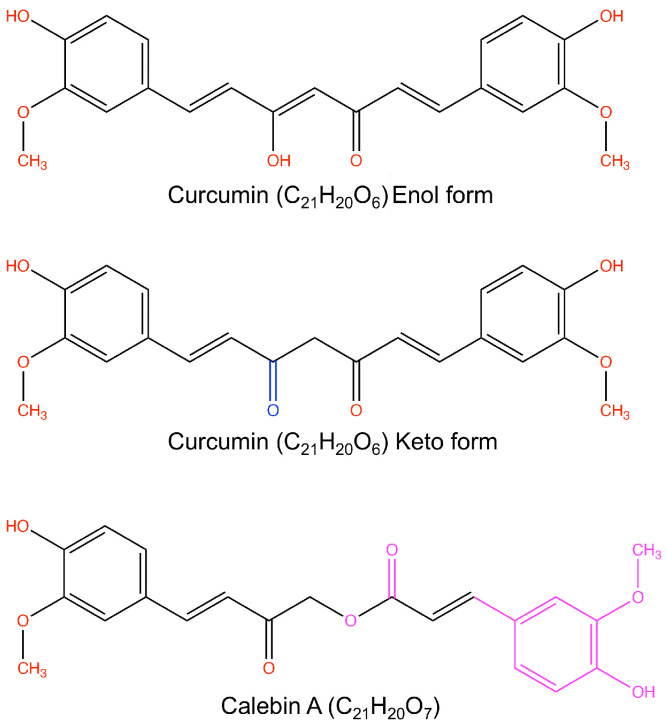
Structures of curcumin and calebin-A. Curcumin has a stable keto form and an unstable enol form. Calebin-A is an analog of curcumin and has a ferulic acid ester bond. The structure was drawn using a ChemDraw v22.0 software.

**Figure 2 ijms-25-03507-f002:**
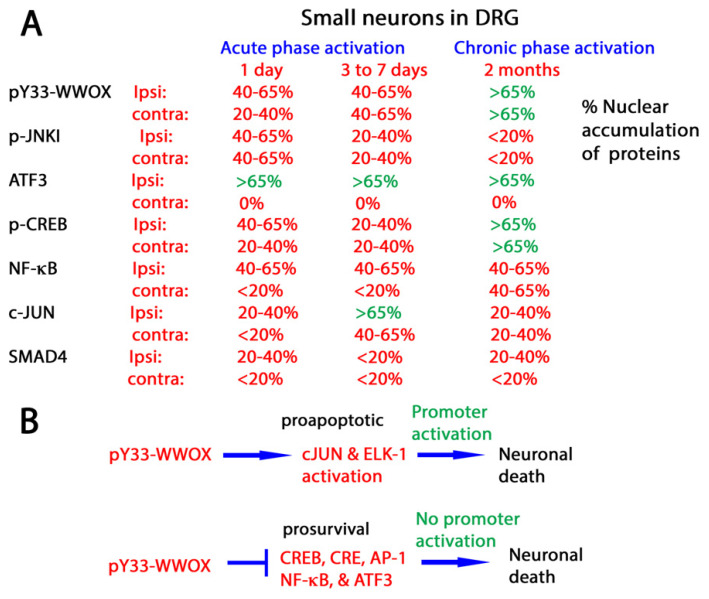
Nuclear accumulation of pY33-WWOX and transcription factors in small neurons in DRG during sciatic nerve dissection. (**A**) The extent of nuclear accumulation of an indicated protein is shown. Ipsi: ipsilateral; Contra: contralateral. (**B**) pY33-WWOX binds and blocks the function of JNK1, NF-κB, CREB, and AP-1 to cause neuronal death. pY33-WWOX also enhances the promoter activation driven by c-JUN and ELK-1 to induce neuronal apoptosis. Binding of pY33-WWOX with ATF3 is unknown. Our data were tabulated and summarized from reference [50].

**Figure 3 ijms-25-03507-f003:**
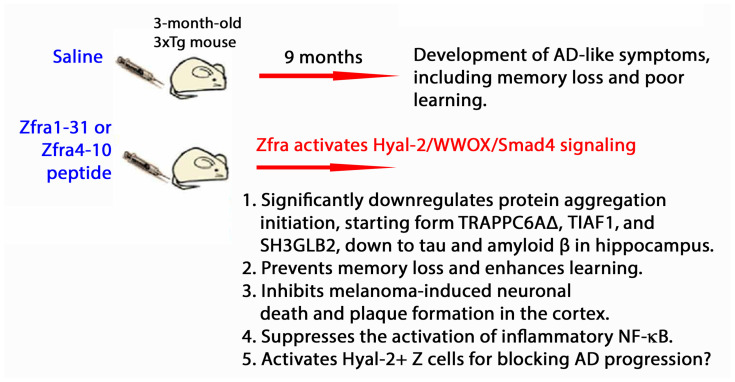
Zfra peptide prevents AD progression and mitigates AD-like symptoms in 3xTg mice. Zfra peptide is potent in mitigating AD-like symptoms in 9-month-old 3xTg mice [71] and prevents the age-dependent memory loss and reduced capability in learning [72].

**Figure 4 ijms-25-03507-f004:**
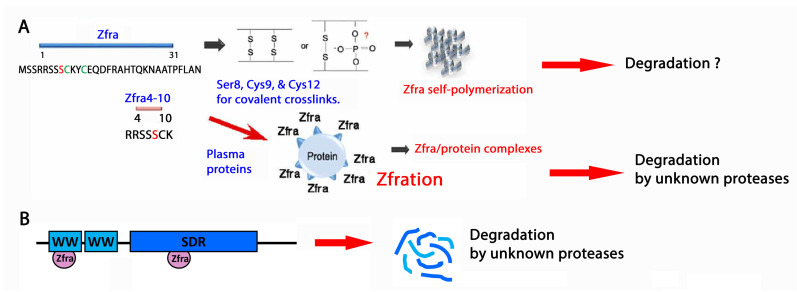
Zfra polymerization and protein zfration for degradation. (**A**) Zfra1-31 and 4-10 peptides can undergo self-polymerization or conjugate with cytosolic or plasma proteins (known as zfration). Cys9 and Cys12 are involved. Participation of Ser8 is being investigated. Degradation of the protein complexes occurs via an unknown mechanism. Known proteinase inhibitors and a proteasome inhibitor MG-132 cannot block protein complex degradation. (**B**) Zfra binds WW1 covalently. How Zfra binds SDR domain is still being determined. The degradation of Zfra/WWOX is not caused by ubiquitination/proteasomes or known proteases.

**Figure 5 ijms-25-03507-f005:**
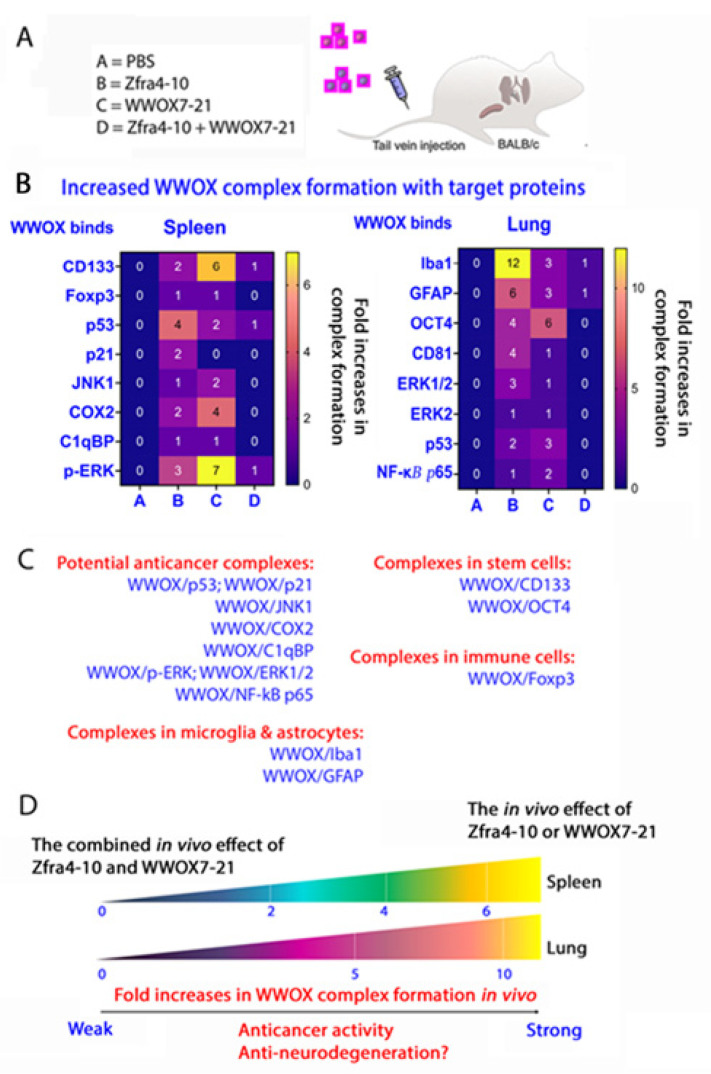
Zfra4-10 or WWOX7-21 upregulates the complex formation of WWOX with target proteins in vivo. (**A**) BALB/c mice received tail vein injections of Zfra4-10 and/or WWOX7-21 peptides, followed by inoculation with 4T1 breast cancer cells two weeks later and sacrifice 18 days later. By co-immunoprecipitation, increased binding of WWOX with C1qBP and other indicated proteins was shown in the spleen and lung. These complexes are present in indicated cells and exosomes. No induction of complex formation was shown when mice received the mixture of Zfra4-10 and/or WWOX7-21 peptides. Functionally, WWOX and p53 may synergistically induce apoptosis [30,48,49]. JNK1 counteracts WWOX function in inducing apoptosis [48]. The binding of WWOX with JNK1 and ERK leads to inhibition of these enzymes’ activity in hyperphosphorylating tau [23]. (**B**) Overall, when endogenous WWOX binds strongly with interacting proteins, the extent of cancer growth is dramatically reduced in vivo. (**C**) The functional properties of WWOX and target protein complexes are shown. (**D**) Zfra4-10 or WWOX7-21 induces anti-cancer responses, which is due to the increased binding strength of pY33-WWOX with its partner proteins, which confers anti-cancer activity in vivo [68].

**Figure 6 ijms-25-03507-f006:**
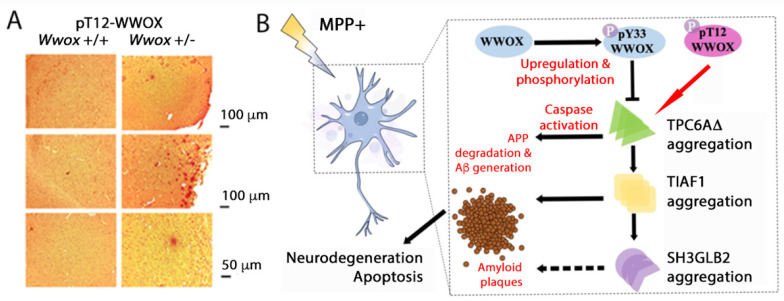
pT12-WWOX as aggregates may initiate the cascade of protein aggregation, leading to neurodegeneration. Identification of pT12-WWOX as aggregates in the brain cortices of 11-month-old heterozygous *Wwox* mice. (**A**) Presence of pT12-WWOX aggregates is shown in the cortices of heterozygous *Wwox* mice (see the red punctate). No pY287-, pY33- and pT12-WWOX aggregates were found in either the wild-type or the heterozygous *Wwox* mice. (**B**) A schematic graph for the action of pY33- and pT12-WWOX, which modulate the protein aggregation cascade of TPC6AΔ, TIAF1, and SH3GLB2. Stimulation of cells with a neurotoxin MPP^+^ causes neuronal apoptosis. Aggregation of TPC6AΔ, TIAF1, and SH3GLB2 occurs during MPP^+^-mediated cell death, suggesting that TPC6AΔ is a common initiator for AD and PD [72].

**Figure 7 ijms-25-03507-f007:**
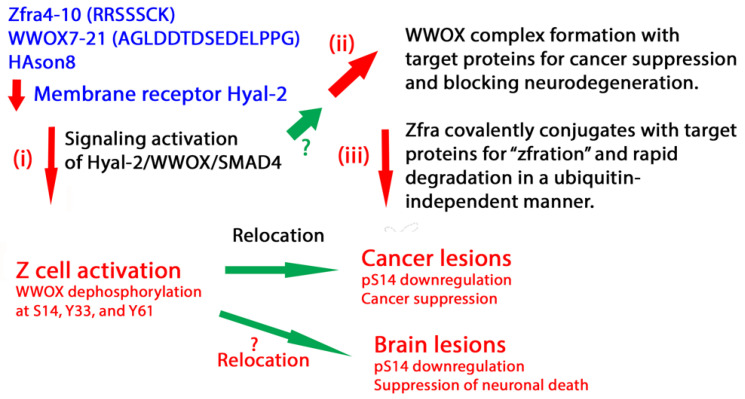
Agonists for activating Hyal-2/WWOX/Smad4 signaling. Suppression of neurodegeneration and cancer growth by Zfra4-10, WWOX7-21, or HAson8 involves (**i**) activation of Z cells [26], which requires dephosphorylation of pS14, pY33, and pY61 in WWOX, (**ii**) enhanced complex formation of intracellular WWOX with binding proteins; and (**iii**) rapid degradation of Zfra-conjugated proteins.

## Data Availability

Not available.

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
