# Peer review of "Zfra Overrides WWOX in Suppressing the Progression of Neurodegeneration"

_ijms, 2024, doi:10.3390/ijms25063507_

Round 1
Reviewer 1 Report
Comments and Suggestions for Authors
Dear Authors, a very informative and comprehensive review on Zfra, WWOX, cancer, and neurodegeneration. I believe the paper can be processed further after appropriate revisions are incorporated. Thus, please answer or consider the following:
(1) It would be helpful to describe what Z cells are on first use, of course excluding Abstract where such information will increase the word count too much.
(2) Whenever you write some examples and end up with “and others”, I would suggest writing delete it and leaving “such as” before examples. For instance, when writing “potential development of multiple diseases such as cancer, neurodegeneration, and others”, I would add maybe a third example except for cancer and neurodegeneration and write “potential development of multiple diseases such as cancer, neurodegeneration, and [third example]”.
(3) I would suggest changing “kill cancer” to “eliminate cancer cells” whenever possible.
(4) Line 52: what is the purpose of writing “desired” between “small” and “peptide”?
(5) The sentences between lines 50 and 66 (introductory part and three points outlining the paper) require improvements. At first, in line 59 you mention that you will describe “molecular signaling and cell fates” while they are above that part (lines 32-49). I think it might be better to rewrite the lines 50-66 in a way that would eliminate repetitions. How about shortening the part in lines 50-58, and then writing what is the general workflow of this review, and not what will be described below? Through that, you will be able to include “molecular signaling and cell fates” even though it is located above that part. Moreover, I would stick with impersonal writing in three points in lines 59-66. Try to make a brief version of current points and delete repeating “we will”. Moreover, in the first point (lines 59-61), please mention that these aspects are not written in the context of Zfra or WWOX but rather a more general approach is practiced here.
(6) Lines 78-79: change “correlate, in part,” to “partly correlate”.
(7) Line 131: consider changing “review articles in the literature for WWOX” to “review articles related to this gene”. Moreover, please remember to italicize the “WWOX” symbol if you refer to gene and not protein (for example in line 148).
(8) Line 134: consider changing “is largely unknown” to “not entirely delineated”.
(9) It would be good to have WW domains abbreviated as WW1 and WW2 to facilitate reading. I think the first mention is in section 4.1. How about adding “abbreviated WW1 and WW2, respectively” to the end of a sentence in line 138? Once abbreviated, you can easily improve the clarity of subsequent sentences.
(10) Line 140: before “PPXY” and “PPPY” abbreviations, please mention that they denote proline-rich motifs.
(11) Line 141-142: change “The event includes” to “Such events include”.
(12) Line 143-144: although the function of the WW2 domain is indeed not entirely known, you can mention that it has some stabilizing properties for WW1. Add relevant citations from the literature.
(13) Line 148-149: optionally, you can add the symbol of a common fragile site related to WWOX (i.e., FRA16D) and mention that it is the second most frequent site of its kind.
(14) For section 4.2, you might add the information that WWOX does not fit Knudson's two-hit hypothesis of tumorigenesis.
(15) Line 166: is “nuclear translocation” correct in this part or it should be “nucleolar translocation”, considering that you mentioned that “TPC6A travels from the mitochondria to nucleoli”? Is WWOX detached from TPC6A in the nucleus during the journey of TPC6A from the mitochondria to nucleoli?
(16) Line 172: change “which is bad for neurons but good for killing” to “which is unfavorable for neurons but beneficial for eliminating”.
(17) Line 177: add “such as” before “p53”.
(18) Line 178: I would move “in vivo” to the beginning of a sentence to write something like “In vivo data confirms that”.
(19) Line 199: change “causes apoptosis of […] rapidly” to “causes rapid apoptosis of […]”.
(20) Line 205-206: change “induced by prosurvival transcription factors CREB, CRE, and AP-1, respectively and leads to neuronal death eventually” to “induced by prosurvival transcription factors CREB, CRE, and AP-1, leading to neuronal death”.
(21) Line 209: I would delete the reference to Figure 1 since a few lines later there is a summarizing paragraph when the same figure is mentioned again.
(22) Line 217-218: combine two short sentences at the end of a paragraph using “whereas”. Moreover, please standardize the use of “ELK-1” or “Elk-1”.
(23) Line 219-228: the entire section 4.6 is a repetition of section 4.2! Please delete one of them.
(24) Figure 1: “DRG” should be explained (dorsal root ganglia?) in the figure or main text, depending on where it was used at first. As for the figure’s legend, I would move all referencing to citation no. 50 at the end of the description. Moreover, write only that “data adapted from reference 50” and move it out of square brackets.
(25) Line 239-240: by “stronger”, you referred to a prevalence or affinity of binding?
(26) Line 254: I would delete the sentence “Without pre-activation, Z cells fail to kill cancer cells”. Is the same message included in the sentence “Non-activated Z cells cannot kill cancer cells […]” (line 250)?
(27) For each section, the last “summarizing” paragraph is frequently a repetition of the same sentences from previous paragraphs. Could you please paraphrase it whenever possible?
(28) Line 283: “respectively” is redundant here.
(29) Line 288: would “baseline” be better than “basal”?
(30) Line 296: in my opinion, the mention of Figure 3 is too early in the text since in the same section there is a description of Figure 2 whereas Figure 2 itself is not yet visualized till the end of section 5.2 (section 6.1 for Figure 3).
(31) Line 301: would it be advantageous to mention that the pS14 modification of WWOX disables its function?
(32) Figure 2: numbers in subfigure B (heatmaps) must be enlarged. Moreover, there is a typo in the bottom right corner: “anticancer actviity”. Lastly, consider changing “which confers resistance to cancer” to “which confers anti-cancer activity” in the last sentence of the figure’s description.
(33) Line 325: “Thus far, no reports demonstrate that WWOX becomes aggregated in the brain.” <-- What about the sentence in line 303, i.e., “pS14-WWOX is accumulated in the lesions of cancers and AD brains”?
(34) Line 346-347: Is the repetition of Figure 3A necessary in two sentences next to each other?
(35) Line 350: there should be a mention of properties of TRAPPC6A-delta on first use, whereas it is found in lines 364-365.
(36) Line 351: change “then” to “and followed by”.
(37) Line 366: “PD” abbreviation was explained earlier.
(38) Figure 3: would it be better to direct a solid black arrow from “pT12 WWOX” at green triangles instead of the adjacent text? Moreover, there is a typo “prootein” in line 382. Explain “MPP” on first use (1-methyl-4-phenylpyridinium?). I would also delete the repetition of citation 79 from the description and just put it once at the end of the figure legend, just like you did with Figure 1.
(39) Line 400: would “inactivating” be better than “defective” next to “phosphorylation”?
(40) Line 438: Does inter- and intra- molecular interaction refer to both WW and SDR domains, if there is only one SDR domain intramolecularly?
(41) Line 440: add a hyphen between “membrane” and “localized”.
(42) Line 444: “to killing” is redundant.
(43) Line 479: using a hyphen between “these” and “treated” is uncommon. Is “these” necessary in the sentence?
(44) Line 509: putting “HAson8” in brackets might be necessary.
(45) Line 512: I think that mentioning Figure 4 is too early considering it is placed in section 11 while the part to which I am referring is the beginning of section 10. Moreover, the mention of Figure 4 is correctly placed and described in section 11 (lines 541-554 with routes i, ii, and iii).
(46) Line 515: avoid repetition of “is unknown” by, e.g., “is also yet to be revealed”.
(47) Line 533: would “which” be better than “that”? Also, a comma might be necessary.
(48) Line 572-573: to avoid repetition of “whether […] remains to be established”, one can use “same in terms of” or equivalent.
(49) Figure 4: line 580 – are data on HAson8 only based on melanoma? Moreover, what is the meaning “[see preliminary data in Section 3 for Research Strategy and Feasibility” in lines 582-583?
(50) Line 608: I think “Not applicable” without quotation marks should be enough but please double-check with the Editorial Office.
Comments on the Quality of English LanguageModerate editing of English language required, some of examples were I mentioned in my peer-review report.
Author Response
Dear Reviewer:
We appreciate your great enthusiasm and outstanding contributions and comments. In response to your comments, we have now addressed point-by-point to your critiques as follows:
Q1: It would be helpful to describe what Z cells are on first use, of course excluding Abstract where such information will increase the word count too much.
Answer: Z cells were originally described in the Section 5.1. As requested, few words have now been added (Line 237 to 245). Z cell was named because Zfra activates this cell type in the spleen [70,71]. Z cell was originally isolated from the T/B cell-deficient NOD-SCID mice. Z cell has a similar size as that of T and B cells. We have determined the gene expression profile of naïve Z cell from NOD-SCID mice (see GEO database Accession: GSE98409, ID: 200098409). Z cell is a new type of lymphoid cells. Zfra-activated Z cells are capable of causing cancer cell death with or without physical contact [70], and are able to block neurodegeneration, including Alzheimer’s disease, Parkinson’s disease, and seizures (unpublished).
Q2: Whenever you write some examples and end up with “and others”, I would suggest writing delete it and leaving “such as” before examples. For instance, when writing “potential development of multiple diseases such as cancer, neurodegeneration, and others”, I would add maybe a third example except for cancer and neurodegeneration and write “potential development of multiple diseases such as cancer, neurodegeneration, and [third example]”.
Answer: Thank you for the great points. As requested, we have looked over the manuscript and revised several areas.
Q3: I would suggest changing “kill cancer” to “eliminate cancer cells” whenever possible.
Answer: Thank you. Many needed areas have been fixed. Either eliminate or eradicate has been used.
Q4: Line 52: what is the purpose of writing “desired” between “small” and “peptide”?
Answer: As requested, “desired” has been deleted.
Q5: The sentences between lines 50 and 66 (introductory part and three points outlining the paper) require improvements. At first, in line 59 you mention that you will describe “molecular signaling and cell fates” while they are above that part (lines 32-49). I think it might be better to rewrite the lines 50-66 in a way that would eliminate repetitions. How about shortening the part in lines 50-58, and then writing what is the general workflow of this review, and not what will be described below? Through that, you will be able to include “molecular signaling and cell fates” even though it is located above that part. Moreover, I would stick with impersonal writing in three points in lines 59-66. Try to make a brief version of current points and delete repeating “we will”. Moreover, in the first point (lines 59-61), please mention that these aspects are not written in the context of Zfra or WWOX but rather a more general approach is practiced here.
Answer: Thank you. As requested, the indicated area has been revised as “In this perspective review, we will go through the fundamental concept of chronic inflammation in neurodegenerative diseases and cancer, update the WWOX functional properties in suppressing Alzheimer’s disease (AD) and cancer, and discuss how Zfra peptide regulates AD and cancer via the Hyal-2/WWOX/Smad4 signaling, and Zfra-activated Z cells in controlling disease progression.” (Line 50-54)
Q6: Lines 78-79: change “correlate, in part,” to “partly correlate”.
Answer: Fixed. (Line 66)
Q7: Line 131: consider changing “review articles in the literature for WWOX” to “review articles related to this gene”. Moreover, please remember to italicize the “WWOX” symbol if you refer to gene and not protein (for example in line 148).
Answer: Fixed. WWOX gene has been italicized in many needed areas. (Line 124)
Q8: Line 134: consider changing “is largely unknown” to “not entirely delineated”.
Answer: Fixed. (Line 127)
Q9: It would be good to have WW domains abbreviated as WW1 and WW2 to facilitate reading. I think the first mention is in section 4.1. How about adding “abbreviated WW1 and WW2, respectively” to the end of a sentence in line 138? Once abbreviated, you can easily improve the clarity of subsequent sentences.
Answer: Fixed. (Section 4.1)
Q10: Line 140: before “PPXY” and “PPPY” abbreviations, please mention that they denote proline-rich motifs.
Answer: Fixed. (Line 133)
Q11: Line 141-142: change “The event includes” to “Such events include”.
Answer: Fixed. (Line 135)
Q12: Line 143-144: although the function of the WW2 domain is indeed not entirely known, you can mention that it has some stabilizing properties for WW1. Add relevant citations from the literature.
Answer: Fixed as “The second WW domain has one tryptophan, whose function has not been elucidated. However, it has been proposed that WW2 teams up with WW1 to maintain an appropriate tertiary conformation that affects the function of WW1 in protein/protein binding [36].” (Line 136-138)
Q13: Line 148-149: optionally, you can add the symbol of a common fragile site related to WWOX (i.e., FRA16D) and mention that it is the second most frequent site of its kind.
Answer: Fixed as “The human WWOX gene, encoding the WWOX protein, is located on a common fragile site, i.e., FRA16D — the second most frequent site of its kind [28-32].” (Line 143-144)
Q14: For section 4.2, you might add the information that WWOX does not fit Knudson's two-hit hypothesis of tumorigenesis.
Answer: Fixed as “Human newborns lacking WWOX gene and functional protein suffer severe neural diseases but do not have spontaneous tumor formation, suggesting WWOX does not fit Knudson's two-hit hypothesis of tumorigenesis [31,35,45,46].” (Line 152-155)
Q15: Line 166: is “nuclear translocation” correct in this part or it should be “nucleolar translocation”, considering that you mentioned that “TPC6A travels from the mitochondria to nucleoli”? Is WWOX detached from TPC6A in the nucleus during the journey of TPC6A from the mitochondria to nucleoli?
Answer: Yes, the WWOX/TPC6A complex dissociates in the nuclei. Then, TPC6A travels to the nucleoli, whereas WWOX remains in the nuclei [81,82]. However, WWOX can travel to the nucleoli in certain types of cells. How this occurs is unknown. (Line 165-167)
Q16: Line 172: change “which is bad for neurons but good for killing” to “which is unfavorable for neurons but beneficial for eliminating”.
Answer: Fixed. (Line 173-174)
Q17: Line 177: add “such as” before “p53”.
Answer: Fixed. (Line 178)
Q18: Line 178: I would move “in vivo” to the beginning of a sentence to write something like “In vivo data confirms that”.
Answer: Fixed. (Line 177)
Q19: Line 199: change “causes apoptosis of […] rapidly” to “causes rapid apoptosis of […]”.
Answer: Fixed. (Line 207)
Q20: Line 205-206: change “induced by prosurvival transcription factors CREB, CRE, and AP-1, respectively and leads to neuronal death eventually” to “induced by prosurvival transcription factors CREB, CRE, and AP-1, leading to neuronal death”.
Answer: Fixed. (Line 212-213)
Q21: Line 209: I would delete the reference to Figure 1 since a few lines later there is a summarizing paragraph when the same figure is mentioned again.
Answer: Fixed. Please note that the original Figure 1 is now Figure 2. (Line 192)
Q22: Line 217-218: combine two short sentences at the end of a paragraph using “whereas”. Moreover, please standardize the use of “ELK-1” or “Elk-1”.
Answer: Fixed. (Line 225 and 226)
Q23: Line 219-228: the entire section 4.6 is a repetition of section 4.2! Please delete one of them.
Answer: Our apologies. Deleted.
Q24: Figure 1: “DRG” should be explained (dorsal root ganglia?) in the figure or main text, depending on where it was used at first. As for the figure’s legend, I would move all referencing to citation no. 50 at the end of the description. Moreover, write only that “data adapted from reference 50” and move it out of square brackets.
Answer: Fixed. Dorsal root ganglion was first mentioned in Line 184.
Q25: Line 239-240: by “stronger”, you referred to a prevalence or affinity of binding?
Answer: Please note that the introductory sentence “We reported that the stronger endogenous WWOX binds intracellular protein partners, the weaker the cancer cells can grow in vivo [68]” has been moved to section 5.5 (Line 333-334). We used organ lysates for co-immunoprecipitation. We believe there is an increased affinity of binding and prevalence. We will repeat the experiments using tissue sections and examine the binding by antibody FRET imaging or affinity proximity assay. This will detail not only the binding affinity, but also the extent of prevalence in binding among cell types. (Line 349-352)
Q26: Line 254: I would delete the sentence “Without pre-activation, Z cells fail to kill cancer cells”. Is the same message included in the sentence “Non-activated Z cells cannot kill cancer cells […]” (line 250)?
Answer: Fixed. (Line 254-255)
Q27: For each section, the last “summarizing” paragraph is frequently a repetition of the same sentences from previous paragraphs. Could you please paraphrase it whenever possible?
Answer: Fixed. Thanks for mentioning about this. We have looked into the whole article and revised wherever it needs.
Q28: Line 283: “respectively” is redundant here.
Answer: Fixed. (Line 337)
Q29: Line 288: would “baseline” be better than “basal”?
Answer: Ok. We changed it to baseline. (Line 342)
Q30: Line 296: in my opinion, the mention of Figure 3 is too early in the text since in the same section there is a description of Figure 2 whereas Figure 2 itself is not yet visualized till the end of section 5.2 (section 6.1 for Figure 3).
Answer: The mention of the original Figure 3 (now Figure 6) has been removed to the end of the new section 6.1.
Q31: Line 301: would it be advantageous to mention that the pS14 modification of WWOX disables its function?
Answer: WWOX has two faces. pY33-WWOX supports normal physiology. Under stress conditions, overexpressed pY33-WWOX induces apoptosis of normal cells, neurons and cancer cells [68,70,71]. When pY33-WWOX is dephosphorylated, S14 phosphorylation is increased. pS14-WWOX supports neurodegeneration and cancer growth. Nonetheless, pS14-WWOX is needed for T/B cell differentiation [45]. The induced T/B cells are able to block bacterial infection [unpublished]. It is not appropriate to consider pS14-WWOX as a disabled form. We have added few lines at the end of section 5.6. (Line 366-368).
Q32: Figure 2: numbers in subfigure B (heatmaps) must be enlarged. Moreover, there is a typo in the bottom right corner: “anticancer actviity”. Lastly, consider changing “which confers resistance to cancer” to “which confers anti-cancer activity” in the last sentence of the figure’s description.
Answer: Fixed. The original Figure 2 is now Figure 5. As requested, the Figure 5 has been revised, the entire figure enlarged, and typo corrected (below Line 369).
Q33: Line 325: “Thus far, no reports demonstrate that WWOX becomes aggregated in the brain.” <-- What about the sentence in line 303, i.e., “pS14-WWOX is accumulated in the lesions of cancers and AD brains”?
Answer: pS14-WWOX is not an aggregated form even though its function in promoting diseases [70,71]. pS14-WWOX is accumulated in the lesions of cancers and AD brains and these proteins are not aggregated. Same with pY33- and pY287-WWOX. We have determined this by both reducing and non-reducing SDS-PAGE and immunohistochemistry. We will confirm the status by filter retardation assay. Yet, pT12-WWOX is an aggregated form. We suspect that pT12/S14-WWOX cannot undergo aggregation. T12 and S14 can counteract each other in controlling mitochondrial membrane potential (unpublished).
Q34: Line 346-347: Is the repetition of Figure 3A necessary in two sentences next to each other?
Answer: Fixed as “The sizes of pT12-WWOX aggregates are about 30 to 60 μm in diameter, which are formed most likely at an earlier age in mice or the middle-aged humans.” The latter “Figure 3A” is removed.
Q35: Line 350: there should be a mention of properties of TRAPPC6A-delta on first use, whereas it is found in lines 364-365.
Answer: Fixed (Line 411). The sentence “Compared to the wild-type, TPC6AD isoform has an internal deletion of 14 amino acids in the N-terminus [45]” has been moved up to the end of the first paragraph of section 6.1. (Line 415-416)
Q36: Line 351: change “then” to “and followed by”.
Answer: Fixed. (Line 411-412)
Q37: Line 366: “PD” abbreviation was explained earlier.
Answer: Fixed. (Line 425)
Q38: Figure 3: would it be better to direct a solid black arrow from “pT12 WWOX” at green triangles instead of the adjacent text? Moreover, there is a typo “prootein” in line 382. Explain “MPP” on first use (1-methyl-4-phenylpyridinium?). I would also delete the repetition of citation 79 from the description and just put it once at the end of the figure legend, just like you did with Figure 1.
Answer: Graph fixed (below Line 425). The original Figure 3 is now Figure 6. Typo (Line 442).
Q39: Line 400: would “inactivating” be better than “defective” next to “phosphorylation”?
Answer: The whole sentence has been revised “A balanced phosphorylation among T12, S14, Y33 and probably Y34 in the WW1 is critical to limit disease manifestation [70,71]” (Line 460-461)
Q40: Line 438: Does inter- and intra- molecular interaction refer to both WW and SDR domains, if there is only one SDR domain intramolecularly?
Answer: Yes. SDR domain in WWOX can bind another SDR in another WWOX. That is, intermolecular associations have the following combinations: WW1/WW1, SDR/SDR, and WW1/SDR. The intramolecular association has WW1/SDR only [57].
Q41: Line 440: add a hyphen between “membrane” and “localized”.
Answer: Fixed. (Line 502)
Q42: Line 444: “to killing” is redundant.
Answer: Fixed as “Treatment of breast 4T1 cancer cells with antibody against the WWOX7-21 epitope makes the cells susceptible to ceritinib, UV irradiation, and many chemotherapeutic drugs [73].” (Line 505-506)
Q43: Line 479: using a hyphen between “these” and “treated” is uncommon. Is “these” necessary in the sentence?
Answer: “these” removed. (Line 541)
Q44: Line 509: putting “HAson8” in brackets might be necessary.
Answer: Fixed. (Line 571)
Q45: Line 512: I think that mentioning Figure 4 is too early considering it is placed in section 11 while the part to which I am referring is the beginning of section 10. Moreover, the mention of Figure 4 is correctly placed and described in section 11 (lines 541-554 with routes i, ii, and iii).
Answer: Fixed. The original Figure 4 is now Figure 7. (Line 574, 608)
Q46: Line 515: avoid repetition of “is unknown” by, e.g., “is also yet to be revealed”.
Answer: Fixed. (Line 571)
Q47: Line 533: would “which” be better than “that”? Also, a comma might be necessary.
Answer: Fixed. (Line 578)
Q48: Line 572-573: to avoid repetition of “whether […] remains to be established”, one can use “same in terms of” or equivalent.
Answer: Fixed as “What remains to be established are the scenario and molecular mechanisms for the induction of TPC6ADpolymerization by pT12-WWOX aggregates and the degradation of pT12-WWOX aggregates by Zfra”. (Line 643-646)
Q49: Figure 4: line 580 – are data on HAson8 only based on melanoma? Moreover, what is the meaning “[see preliminary data in Section 3 for Research Strategy and Feasibility” in lines 582-583?
Answer: In addition to melanoma, HAson8 has been tested in breast, lung, prostate, and skin cancer cells, as well as neuroblastoma, growth in vivo (unpublished). HAson8 also retards the progression of Alzheimer’s disease, Parkinson’s disease, and seizure (manuscript submitted). We have deleted “[see preliminary data in Section 3 for Research Strategy and Feasibility” in lines 582-583?”. Our apologies for the mistakes. (Line 623)
Q50: Line 608: I think “Not applicable” without quotation marks should be enough but please double-check with the Editorial Office.
Answer: Fixed. (Line 666)
Additional changes:
- 1. At the request of Reviewer 2, we have added three new figures:
Figure 1. Structures of curcumin and calebin-A;
Figure 3. Zfra peptide prevents AD progression and mitigates AD-like symptoms in 3xTg mice;
Figure 4. Zfra polymerization and protein zfration for degradation.
- At the request of the editorial, we have removed “cancer” from the title. Also, many areas dealing with cancer have been toned down to fit the agenda of the specific issue in IJMS.

Reviewer 2 Report
Comments and Suggestions for Authors
This paper aims to prove that 31-amino-acid Zfra protein blocks cancer growth and neurodegeneration. The authors focused on whether a single stimulating reagent can induce a specific signaling pathway that limits both the initiation and progression of both cancer and neurodegeneration and they explained the WWOX-binding peptide Zfra for preventing and treating cancer and AD, they also demonstrated the relationship between the chronic inflammation, cancer and neurodegenerative diseases.
The manuscript is written comprehensively enough to be understandable despite of the complexity of the subject; the authors addressed this aim by demonstrating the workflow of the designed study strategy starting by talking about WWOX functional properties, and its role in suppressing cancer and AD, particularly dealing with p53 and WWOX in AD. Then they described the role of Zfra in regulating cancer and AD, and update the functional relationship with epitopes WWOX7-21 and WWOX286-299 in neuronal migration, and the Hyal-2/WWOX/Smad4 signaling.
The paper stated the purpose, discussion and global implication are clearly stated and consistent with the rest of the manuscript; authors provided the required enough information in their discussion by using a good number of important articles talked about the subject.
The authors addressed their hypothesis and opinion in a reproducible way, they used enough number of analyses to prove their results.
The results were presented in a clear way which facilitate in reaching a conclusion elucidates that all the collected data in this study align with the literature previous data and prove the importance of Synthetic Zfra peptides in the treatment of cancer, however they missed some points as follow:
1- The authors should add the chemical structure of Curcumin and its analogue Calebin-A.
2- This review explained how Zfra binds WW domain-containing oxidoreductase (WWOX) to both N- and C-terminal, and leads to accelerated WWOX degradation, so the figure that explains this binding and the mechanism of action should be added.
3- Two reviews could be useful to support the subject:
A- Su, W.-P.; Wang, W.Chang, J.-Y.; Ho, P.-C.; Liu, T.-Y.; Wen, K.-Y.; Kuo, H.-L.; Chen, Y.-J.; Huang, S.-S.; Subhan, D.; et al. Therapeutic Zfra4-10 or WWOX7-21 Peptide Induces Complex Formation of WWOX with Selective Protein Targets in Organs that Leads to Cancer Suppression and Spleen Cytotoxic Memory Z Cell Activation In Vivo. Cancers 2020, 12, 2189. https://doi.org/10.3390/cancers12082189
B- Lin YH, Shih YH, Yap YV, Chen YW, Kuo HL, Liu TY, Hsu LJ, Kuo YM, Chang NS. Zfra Inhibits the TRAPPC6AΔ-Initiated Pathway of Neurodegeneration. Int J Mol Sci. 2022 Nov 22;23(23):14510. doi: 10.3390/ijms232314510. PMID: 36498839; PMCID: PMC9739312.
No plagiarism has been detected.
The abbreviations should be explained at the first place they are mentioned.
In vitro, in vivo, et al.: should be written in italic.
References: The authors followed the journal guidelines for some references.
Author Response
Dear Reviewer:
First of all, thank you so much for your enthusiastic efforts and helpful comments. We have now answered your comments point-by-point as follows:
Q1. The authors should add the chemical structure of Curcumin and its analogue Calebin-A.
Answer: As requested, the chemical structures of Curcumin and its analogue Calebin-A have been added in the new Figure 1. The legend explains the differences between these two chemicals.
Q2. This review explained how Zfra binds WW domain-containing oxidoreductase (WWOX) to both N- and C-terminal, and leads to accelerated WWOX degradation, so the figure that explains this binding and the mechanism of action should be added.
Answer: As requested, we have added a schematic graph showing Zfra binds to the N- and C-termini of WWOX. Based on our published data, we believe that non-phosphorylated Zfra binds Y33 phosphorylated WW1 domain. Zfra binds Y287 phosphorylated SDR domain. Compared to the non-phosphorylated Zfra, pS8-Zfra is not functionally active. This binding may lead to accelerated WWOX degradation. Similarly, Zfra binds many other proteins and the resulting complexes are subjected to degradation by unknown mechanisms, not by ubiquitination/proteasome or by known proteases. The zfra action is known as "zfration". The designed graph is shown in the Figure 4.
Q3. Two reviews could be useful to support the subject:
A- Su, W.-P.; Wang, W.Chang, J.-Y.; Ho, P.-C.; Liu, T.-Y.; Wen, K.-Y.; Kuo, H.-L.; Chen, Y.-J.; Huang, S.-S.; Subhan, D.; et al. Therapeutic Zfra4-10 or WWOX7-21 Peptide Induces Complex Formation of WWOX with Selective Protein Targets in Organs that Leads to Cancer Suppression and Spleen Cytotoxic Memory Z Cell Activation In Vivo. Cancers 2020, 12, 2189. https://doi.org/10.3390/cancers12082189
B- Lin YH, Shih YH, Yap YV, Chen YW, Kuo HL, Liu TY, Hsu LJ, Kuo YM, Chang NS. Zfra Inhibits the TRAPPC6AΔ-Initiated Pathway of Neurodegeneration. Int J Mol Sci. 2022 Nov 22;23(23):14510. doi: 10.3390/ijms232314510. PMID: 36498839; PMCID: PMC9739312.
Answer: We have mentioned both articles in the manuscript, wherever they are needed. Failure in Zfra-mediated protein degradation may lead to neurodegeneration (71). Zfra can bind amyloid beta aggregates without causing degradation, or both proteins colocalize without binding each other.
Additional changes:
- 1. As requested, we have added three new figures:
Figure 1. Structures of curcumin and calebin-A;
Figure 3. Zfra peptide prevents AD progression and mitigates AD-like symptoms in 3xTg mice;
Figure 4. Zfra polymerization and protein zfration for degradation.
- 2. At the request of the editorial, we have removed “cancer” from the title, and have toned down many areas dealing with cancer to fit the agenda of the specific issue in IJMS.
